# Mechanisms of Blood–Brain Barrier Dysfunction in Traumatic Brain Injury

**DOI:** 10.3390/ijms21093344

**Published:** 2020-05-08

**Authors:** Alison Cash, Michelle H. Theus

**Affiliations:** 1The Department of Biomedical Sciences and Pathobiology, Virginia-Maryland College of Veterinary Medicine, Blacksburg, VA 24061, USA; amcash3@vt.edu; 2The Center for Regenerative Medicine, Virginia-Maryland College of Veterinary Medicine, Blacksburg, VA 24061, USA

**Keywords:** Blood–brain barrier disruption, TBI, endothelial cells, vascular–astrocyte coupling, aquaporin, edema

## Abstract

Traumatic brain injuries (TBIs) account for the majority of injury-related deaths in the United States with roughly two million TBIs occurring annually. Due to the spectrum of severity and heterogeneity in TBIs, investigation into the secondary injury is necessary in order to formulate an effective treatment. A mechanical consequence of trauma involves dysregulation of the blood–brain barrier (BBB) which contributes to secondary injury and exposure of peripheral components to the brain parenchyma. Recent studies have shed light on the mechanisms of BBB breakdown in TBI including novel intracellular signaling and cell–cell interactions within the BBB niche. The current review provides an overview of the BBB, novel detection methods for disruption, and the cellular and molecular mechanisms implicated in regulating its stability following TBI.

## 1. Traumatic Brain Injury

### 1.1. Overview of Traumatic Brain Injury

Traumatic brain injury (TBI) is the leading cause of injury-related death and disability in the United States (CDC; TBI Surveillance System). TBI results from external forces that hinder normal brain function. These range from mild, to moderate, to severe. The metrics for classification of injury severity include a variety of factors including level of consciousness, amnesia status, and neuroimaging modalities. The commonly-used Glasgow Coma Scale (GCS) evaluates consciousness on a scale of 1–15 as a measurement of eye opening, verbal, and motor responses [1,2]. As defined by the Veterans Health Administration and the Department of Defense, mild injuries are predominantly sport and concussion-related injuries characterized by a brief alteration of consciousness. Whereas, moderate to severe diagnoses are harder to classify since they involve criteria that are more diverse. These are commonly based on loss of consciousness for greater than 30 min, at least a day of amnesia, and scores below 13 on the GCS [3]. Additionally, severe injuries are associated with serious physical and mental impairment and a GCS score of ≥ 8 [2]. While GCS is a valuable tool used to index TBI severity, it relies on observable outcomes and, therefore, a growing need exists to improve the physiological tools to complement the traditional classification system. Other, less common metrics, include time to follow command, loss of consciousness, post-traumatic amnesia [4], and the Abbreviated Injury Score [4,5,6]. While these approaches focus on clinical neurological scoring, additional measures such as blood–brain barrier (BBB) assessment may aide in diagnosis and long-term management of TBI. However, such strategies will require the use of new imaging modalities and detection of circulating brain-derived molecules. Mechanical insult to the brain causes immediate tissue deformation, shear stress, and damage to surrounding blood vessels [7,8]. Following the primary impact, BBB disruption contributes to tissue damage, subsequent edema, inflammation, and neural dysfunction [1,9]. BBB breakdown has long-lasting effects and is associated with neurodegeneration or other comorbidities such as Alzheimer’s disease and epilepsy [10,11,12,13,14], making it a crucial target for treatment.

### 1.2. Blood–Brain Barrier Disruption in TBI

BBB disruption is a known consequence of TBI and is associated with poorer outcomes [9]. BBB disruption is an early event, occurring within hours following injury [15,16,17,18,19] but can persist for years [19,20,21], and is considered a major risk factor for high mortality and morbidity (Table 1). Extravasation of the serum protein, fibrinogen (FBG), and immunoglobulin G (IgG), both markers for BBB disruption, were observed in the brains of human patients that died in the acute phase following TBI, as well as in those that survived at least a year [19]. Additional findings showed increased fibrinogen in the human brain 6–72 h following severe TBI [22]. Early restoration of BBB integrity may also aid in preventing the sequelae of other co-morbidities associated with TBI such as post-traumatic epilepsy (PTE) and neurodegenerative disease [9]. It has been shown that focal motor seizures occur immediately after osmotically-induced BBB disruption in human patients, contralateral to the site of disruption which was confirmed in porcine models [23]. Persistent BBB breakdown in PTE patients may contribute to its pathogenesis [24]. Moreover, fluorescein isothiocyanate (FITC)-labelled plasma component Abeta42, implicated in Alzheimer’s disease (AD), crossed the BBB and was found to be accumulated in the brains of AD patients [25,26]. Emphasis on the multifaceted role of the BBB in acute and chronic outcomes following TBI will help advance new detection methods and prognostic and treatment strategies.

Disruption of two key biological processes contributes to BBB dysfunction. The first involves an increase in paracellular transport indicated by a loss of tight junction (TJ) proteins, allowing passage of molecules that would usually be restricted [9,39,40]. This includes an influx of immune cells such as neutrophils that can exacerbate the inflammatory response. The second process occurs due to an increase in transcytosis across the endothelial cell (EC), leading to transport of larger molecules and serum proteins, such as albumin, which are normally restricted from entering the brain [9,39,40]. In addition, a major consequence of BBB disruption is cerebral edema or swelling due to excess water accumulation in the brain. Generation of cerebral edema following injury is heterogeneous, based on severity and the brain region most affected. Two types of edema can occur following brain injury, cytotoxic and vasogenic. Following BBB breakdown, vasogenic edema results following fluid accumulation in the peri-vascular space leading to changes in cerebral blood flow (CBF), and increased intracranial pressure (ICP) [41,42]. On the other hand, cytotoxic edema is caused by activation of ion channels that drive influx of water into the intracellular space of various cell types resulting in further disruption to the BBB [42,43,44,45]. Elevated ICP and changes in CBF that are not immediately controlled can result in irreversible tissue damage and cell death, which contributes to the high mortality in severe cases of TBI [42,45].

### 1.3. Animal Models in the Study of BBB Disruption Following TBI

Numerous animal models have demonstrated enhanced BBB permeability within hours following TBI [46,47,48] (Table 1). The BBB is disrupted as early as 1-6 h following fluid-percussion injury in rabbits, 1 h following murine controlled cortical impact (CCI), 1 h following severe TBI in swine, and immediately following closed head injury in rats [17,18,22,28,46]. A timeline of BBB permeability at the site of injury demonstrates extravasation of IgG which peaked 24-72 h, remained increased at 7 days post-injury, then was restored at 60–180 days following TBI in rats [30]. Employment of large-animal models, which more closely resemble human TBI, have also contributed to our understanding of BBB breakdown (Table 1). Immunohistochemical (IHC) analysis of serum FBG in the swine showed staining in the cortex indicating significant BBB disruption at 6-72 h following diffuse TBI that overlapped with axonal injury and paralleled similar FBG extravasation in post-mortem human tissue [33]. Dysregulation of the BBB correlates with reduced expression of TJ proteins such as occludin, claudin-5, and zona occludens [22,30,33,47,49]. It is important to note that animal models come with inherent limitations when translating to human TBI, such as differences in anatomy and brain size. While mice and rat studies far exceed other models due to efficiency, they have obvious dissimilarities to humans. Not only are they quadrupeds with vastly different brain sizes, but mice and rats are lissencephalic, whereas, humans are gyrencephalic [50]. Therefore, complementary studies using large animal models such as swine and non-human primates with gyrencephalic brains is warranted. However, these models impose their own constraints such as increased cost, and space and ethical issues. BBB disruption occurs across numerous species following TBI; however, the converging cellular and molecular mechanism(s) regulating this response require further investigation.

### 1.4. Clinical Assessment of BBB Disruption in TBI

Human studies of BBB function require the ability to visualize and detect the dynamic changes that may occur in response to perturbation. Non-invasive imaging techniques represent a highly attractive method to assess and monitor BBB breakdown in patients following TBI. Computed tomography (CT) scans are a non-invasive test that can distinguish acute intracranial pathology such as sites of hemorrhage, tissue swelling, and foreign bodies [51,52]. Hemorrhage, or fluid (blood) accumulation, is observed as regions of abnormal hyperdensity [52]. CT is superior to other imaging in that it is accessible and highly sensitive in the acute stages following injury. It is better able to distinguish clot from brain parenchyma as compared to the magnetic resonance imaging (MRI) [53,54]. However, once the composition of hemorrhage changes in the days following TBI, subjective analysis becomes difficult [55]. Moreover, certain contrast CT scans, such as dynamic contrast-enhanced CT (DCE-CT) use ionizing radiation and an iodine contrast that may cause adverse reactions [56]. The most widely used brain imaging modality is MRI, in particular T1-weighted dynamic contrast-enhanced MRI [17,57,58,59,60,61,62]. This is a sensitive, minimally-invasive method used to quantify the functional integrity of the BBB [57] by evaluating high-intensity acute reperfusion marker (HARM), and is found to be the most influential factor in predicting early BBB injury in stroke [59,60,61] and TBI [17,62]. Importantly, mapping water exchange across the BBB can be achieved using 3D diffusion-prepared or multiple echo time arterial-spin-labeled (ASL) perfusion MRI. This non-invasive approach bypasses the use of exogenous contrast agents and utilizes water exchange rate across the BBB, which may potentially serve as a measure of BBB function [63,64,65].

More advanced imaging modalities such as diffusion tensor imaging (DTI) and susceptibility-weighted imaging (SWI) may be valuable as prognostic tools in evaluating BBB integrity as they provide information regarding water diffusion in multiple spatial directions [66,67]. DTI provides fractional anisotropy (FA) values which indicate the overall directionality of water diffusion [68,69,70,71,72]. DTI can also differentiate between increased diffusivity resulting from vasogenic edema compared to decreased diffusivity shown by cytotoxic edema. This is due to water molecules moving more freely within the extracellular vs. intracellular space, respectively. Increased FA values are seen in human cases 72 h following mild TBI [73], and persistent depressed FA values in human patients weeks to years following mild and severe brain injuries [74,75,76,77]. These changes are associated with poorer outcomes and, therefore, may be a valuable prognostic tool in TBI [73,74,75,76,77]. There is mounting evidence that even in cases of mild concussion, there are areas of focal cerebral microbleeds, which may increase the risk of intracranial hemorrhage and neurological dysfunction [78,79]. These are detectable on magnetic resonance SWI [78,79,80,81]. SWI is hypersensitive compared to other weighted MRI modalities and microbleeds appear as circular hypointense lesions [80,82]. However, disadvantages of MRI are that it is expensive, time-consuming, requires a contrast agent, and may have contraindications such as pacemakers or other metal implants [83,84]. Evaluation of BBB permeability in TBI may also be possible using perfusion CT, as well as, single-photon emission computerized tomography-diethylenetriaminepentaacetic acid (SPECT-DCTA) as seen in stroke [85,86,87,88,89]. Furthermore, DCE-MRI is capable of picking up subtle changes in BBB disruption following stroke [90] and in Alzheimer’s disease [57,91] suggesting its use in TBI may be valuable. Therefore, combinations of MRI or CT may provide the best picture of the BBB pathophysiology in order to assess and monitor its function following TBI or in response to treatment paradigms [87].

Importantly, these imaging techniques are being complimented by evidence-based evaluation of serum biomarkers. Protein biomarkers represent a highly attractive diagnostic tool with the potential to detect substantial changes in BBB disruption that may be predictive of injury severity and outcome (Table 2). Biomarkers of BBB damage may include vascular-related structural proteins such as fragmented TJ proteins (occludin, claudin-5, and zona occludens), increased cerebral spinal fluid (CSF) [92], and increased occludin levels in the blood [93] as seen following ischemic stroke. Plasma albumin and brain-specific proteins such as glial fibrillary acid protein (GFAP), ubiquitin carboxyl-terminal hydrolase isozyme L1 (UCH-L1), and S100 calcium-binding protein B (S100B) are elevated in the serum following TBI which correlates with BBB dysfunction [36,37]. Recent studies combined the use of DTI and serum assessment of S100B and S100B autoantibodies to evaluate low-force head impact and BBB disruption [35]. Furthermore, in a mouse model of TBI, shear stress resulted in the release of extracellular microvesicles (eMVs) containing TJ proteins, specifically occludin, within 24 h following injury from the brain endothelium [94]. Similarly, microRNA (miRNA) alterations in serum and plasma, as a measure of BBB disruption, have recently gained attention in TBI [38,95]. miRNAs are small, endogenous, post-transcriptional gene regulators that play critical roles in various biological processes such as translational suppression or degradation of mRNA, and have also been suggested to contribute to many pathological conditions [95,96]. In human patients, a microarray analysis of 108 microRNAs showed that 52 miRNAs were altered 24 h following severe TBI as compared to healthy individuals [38]. These studies suggest miRNA detection may represent a new diagnostic and prognostic tool for TBI [97,98,99,100]. Additional studies are needed to determine whether these markers meet specificity, sensitivity, and reliability requirements. Improving our understanding of the mechanisms of BBB breakdown will also reveal additional cellular or biochemical targets that are highly sensitive and specific for BBB damage following TBI.

## 2. Blood–Brain Barrier (BBB)

### Overview of BBB

BBB function is critical in maintaining brain homeostasis by suppressing entry of peripheral immune cells and providing nutrient delivery and toxic substance removal, as well as acting as a solute exchange barrier between blood and brain. The BBB provides a semi-permeable barrier separating the circulating blood from the brain environment to regulate molecules that undergo influx and efflux. The BBB is composed of specialized endothelial cells that are fenestrated by transmembrane proteins known as tight junction proteins (occludin and claudins) or junctional adhesion molecules (JAMs) that restrict paracellular permeability [156,157,158,159]. These transmembrane proteins are anchored to the cytoplasmic surface via scaffolding proteins, zonula occludens (ZO) [160]. Select compounds such as small lipid soluble molecules, water, and oxygen passively diffuse across the BBB, whereas, other larger molecules and nutrients such as amino acids, insulin, and plasma proteins require transporters or endocytosis to traverse the BBB. In order for these transporters to mediate homeostasis, assembly within the plasma membrane pore of the endothelial cell (EC) is paramount. The ECs have both a luminal (blood-facing) and abluminal (brain-facing) assembly of transporters that are required for proper BBB function [161].

Endothelial cells that make up the BBB are regulated by surrounding cells such as pericytes, astrocytes, neurons, and microglia in order to maintain ionic balance and homeostasis [162]. They influence ECs directly through secretion of soluble factors or interactions of cell-to-cell contact proteins [163]. Each cell type plays an integral role in the development and maintenance of the BBB. Formation of the murine BBB begins during embryonic day 12 (E12) with sprouting angiogenesis to form a vascular network within the neuroectoderm that is mediated, in part, by a vascular endothelial growth factor (VEGF) gradient [164,165]. At this stage, vessels contain immature barrier properties such as tight junctions, nutrient transporters, and leukocyte adhesion molecules [165]. Pericytes follow vessel sprouting around E19 and ensheath blood vessels to mediate capillary diameter, stability, and blood flow [165,166,167]. Pericytes provide the initial support to ECs, influencing transcellular transport; however, they do not affect TJ integrity [168,169]. Mice born with pericyte deficiencies exhibited increased BBB permeability through higher rates of endothelial transcytosis [170]. Further studies utilizing null mice for platelet-derived growth factor receptor (Pdgfrβ), show reduced pericyte numbers and BBB dysfunction. These findings demonstrate that pericyte coverage of the vasculature is necessary for tight junction formation, vesicle trafficking, and suppression of genes that promote BBB permeability, such as angiopoietin-2 (Angpt2), plasmalemma vesicle-associated protein (Plvap), and leukocyte adhesion molecules [166,171]. Pericytes are also integral as the intermediary between astrocytes and ECs by guiding astrocytic end-feet to the endothelium, creating the necessary polarity and subsequent maturation of the BBB [165].

Following birth, astrocytes proliferate and mature to ensheath ECs with distal projections known as end-feet. Maturation and stability of the BBB occurs through astrocyte release of soluble factors, such as transforming growth factor beta (TGFb), glial-derived neurotrophic factor (GDNF), basic fibroblast growth factor (bFGF), and angiopoietin-1 (Angpt1) which promote BBB integrity [172,173,174,175]. These astrocyte-derived factors are crucial to BBB maturation as confirmed using endothelial cell cultures. ECs co-cultured with astrocytes exhibited a BBB phenotype and improved barrier function, including expression of junctional proteins such as zona occludens (ZO-1), as compared to cultures of ECs alone [176,177,178]. Signaling pathways essential during early development of the BBB include Angpt1/Tie2, Eph/ephrin, and Hedgehog [179,180,181,182] (Table 2). Astrocytes directly regulate endothelial cell expression of tight junction proteins through release of Src-suppressed C kinase substrate (SSeCKS), Angpt1, which in turn, acts on endothelial-derived Tie2 receptor to promote barrier integrity [183,184]. Angpt1 expression has been implicated in barrier maintenance through tyrosine dephosphorylation of occludin, promoting interaction between occludin and ZO-1 [185]. The Sonic-Hedgehog (Shh) pathway is also an important developmental mediator of BBB integrity [182]. Mice devoid of Shh show embryonic lethality, which is associated with decreased expression of claudin-5 and occludin proteins [182]. Furthermore, confirmation of astrocyte-derived Shh on up-regulation of TJ proteins in human endothelial cells was confirmed through in vitro studies [182]. Astrocytes continue to maintain the BBB through control of water and ion gradients via aquaporin 4 (AQP4) and Kir4.1, respectively [165,186]. Various neurological conditions can alter homeostasis within the BBB niche, which may contribute to neural dysfunction.

## 3. Cellular and Molecular Mechanisms Regulating BBB Disruption Following TBI

### 3.1. Neuroinflammation

The secondary injury response including neuroinflammation and glial activation can contribute to disruption of the BBB (Table 2). TBI elicits activation of the endothelium and a neuroinflammatory response within minutes to hours post-injury, which is demonstrated by recruitment and up-regulation of cytokines, chemokines, neutrophils, and other pro-inflammatory mediators [110,187,188,189]. Interestingly, pericytes have recently gained attention as inflammatory mediators of the BBB following injury [190]. However, our understanding of this response is sparse due to difficulties with how quickly pericytes change phenotypes following injury and complications in identifying their structure and function. Innate immune cells also drive neuroinflammation and BBB dysregulation. Microglia, resident innate immune cells, undergo activation by albumin that has entered the brain via transcytosis leading to release of IL-1β, TNF, TGFβ, and MCP-1 [110,111,112,113] which influence BBB permeability and TJ distribution [191,192,193]. Albumin extravasation also causes astrocytes to release matrix metalloproteinases (MMPs) which degrade the basement membrane leading to BBB permeability [114], and increased vasogenic edema following TBI [103]. This is further highlighted by studies showing that suppression of CypA-MMP-9 signaling by apolipoprotein-E (ApoE) regulates BBB integrity following TBI in an isoform-dependent manner [27]. ApoE polymorphisms have gained attention in TBI due to their association with BBB breakdown in cases of deficiency or deletion [194,195], and correlation to late-onset Alzheimer’s disease (ApoE4) [194,196]. This suggests differential ApoE isoform expression may result in divergent patient-specific BBB outcomes following TBI.

Cytokine release by glial cells activates downstream pathways such as Rho/ROCK, PKC, and MAPK, which affect phosphorylation of TJ proteins and contribute to increased paracellular permeability of the BBB [115,116,117,118]. MAPK continues to gain attention for its role in temporal lobe epilepsy and intracerebral hemorrhage (ICH) [119,120]. Transcriptomic analysis of human temporal lobe epilepsy patients showed increased AQP4 expression that may be regulated by the MAPK signaling pathway. When human astrocytes from patients with temporal lobe epilepsy were treated in culture with p-38 MAPK inhibitors, expression of AQP4 was downregulated [119]. Additionally, mice treated with propagermanium, a chemokine CC ligand 2 (CCL2) inhibitor, decreased the expression of MAPK and AQP4 which correlated with a reduction in edema and behavioral deficits [120]. Moreover, studies show monocytes are regulators of BBB permeability through release of oncostatin M and VEGF, which activate glial cells and endothelial cells [121,122,123,124]. Human studies have also shown an increase in malondialdehyde, a byproduct of lipid peroxidation and oxidative stress, immediately following trauma. Oxidative stress is a known disrupter of the BBB and may be the initial catalyst for increased permeability [125,126]. Following mitochondrial damage, reactive oxygen species (ROS) released from astrocytes, microglia, and neurons further activate glial release of cytokines and chemokines [126,127,128,129]. ROS affect downstream pathways that decrease TJ expression, and increase MMPs to enhance paracellular permeability through lipid peroxidation [197,198]. Microglia are also responsible for releasing IL-1 and IL-6, which enhance intracellular adhesion molecule-1 (ICAM-1), P-selectin, and E-selectin expression, thus allowing increased leukocyte adhesion and migration across the brain endothelium to elicit additional peripheral-derived neuroinflammatory responses [118,130,131].

### 3.2. Vascular-Astrocyte Coupling

Communication between endothelial cells and astrocytes has been termed vascular–astrocyte coupling. Recent studies have suggested that the location and distribution of astrocytic end-feet coverage on the microvasculature may influence BBB permeability [104,175,199]. Furthermore, displacement of end-feet from the endothelium results in disruption of the BBB in cases of invading gliomas, as well as, multiple sclerosis [175]. However, recent findings demonstrate that selective loss of astrocytes using laser ablation methods under naïve conditions was not sufficient to increase vascular leakage [200].

Surprisingly, our understanding of vascular-astrocyte coupling in BBB disruption following TBI is limited. Aquaporin 4 (AQP4), a water-channel protein predominantly expressed at the junction of ECs and astrocytic end-feet, maintains ion concentrations and fluid homeostasis, while mediating edema and brain swelling in TBI [101,105,132]. In response to changes in tonicity associated with brain swelling, rat primary cortical astrocytes exhibited re-localization of AQP4 [133]. This perivascular channel is beneficial in water clearance during vasogenic edema but can exacerbate cytotoxic edema [102], therefore, further examination is needed into surface and protein-level expression in pathological conditions. Increased expression of AQP4 is present in rats at 1, 4, and 24 h following TBI [134]. Recent studies have evaluated expression of AQP4 following mild and severe human TBI and report increased expression in tissues and cerebral spinal fluid (CSF) [135,136,201]. Studies show that deletion of AQP4 attenuated BBB disruption, edema, and loss of TJ expression in ischemic conditions [137,202,203]. This supports the notion that up-regulation and redistribution of AQP4 is correlated to BBB disruption in cases of glioblastoma and cerebral ischemia [204,205]. However, the effects of AQP4 deletion on the cerebral vascular may be context dependent. For example, deletion of AQP4 results in a chronic increase in cerebral vascularization in adult mice due to maladaptive water exchange across the BBB in order to maintain CBF [137]. Additional studies using astroglial-conditional *Aqp4* knockout mice further highlights the importance of astrocytic AQP4 in brain water uptake without disruption of barrier function to macromolecules in response to hypoosmotic stress. Moreover, global *Aqp4* knockout mice show reduced expression of perivascular glial scaffolding proteins while BBB function remained intact under normal conditions [138,206].

It remains unclear how these developmentally-driven changes influence the outcome in models of brain injury. Nonetheless, AQP4 remains a potential therapeutic target in the acute and chronic management of BBB disruption, which could influence the onset of other comorbidities. However, additional studies are needed to improve our understanding of how changes in the overall expression and subcellular localization controls BBB function following TBI. Changes in AQP4 subcellular localization, either at the end-foot or mis-localized to other membranes, has also been shown to contribute to BBB dysfunction [106,207,208]. Under certain conditions, modulation of AQP4 expression and its redistribution may be mutually exclusive events [207,209]. For example, when exposed to hypothermic conditions, human primary cortical astrocytes in culture showed increased surface localization without accompanying increases in protein expression level [209]. On the other hand, increased expression and redistribution of AQP4 from the perivascular end-foot to the neuropil was demonstrated in mice that developed PTE following TBI [207].

A primary role of astrocytes is uptake of glutamate through transporters, EAAT1 and EAAT2 [147]. Decreased expression of these transporters is seen in human TBI and may contribute to neurotoxicity [125,148]. Excessive glutamate leads to disruption of the BBB through its activation of NMDA receptors, which enhances vascular permeability and seizures in rats, while NMDA antagonists reduced BBB permeability [149]. Overall, these studies suggest glial-derived factors play an important functional role in BBB homeostasis and TBI-induced disruption. Astrocytes also influence endothelial activity through release of soluble molecules. In particular, MMPs, VEGF, endothelin-1 (ET-1), and glutamate [114,139,140,142,200] released by astrocytes have been linked to BBB disruption. Increased release of MMP-9, an enzyme that degrades the extracellular matrix (ECM), following brain injury has been associated with increased BBB permeability through degradation of TJ proteins, occludin, and claudin-5 [114,210]. Astrocytes also influence the brain endothelium through VEGF signaling. Release of VEGF-A increased BBB disruption through down-regulation of claudin-5 and occludin in a mouse model of cerebral inflammation [141]. VEGF-A interacts with thymidine phosphorylase (TYMP), another astrocyte-derived pro-permeability factor, to promote breakdown through repression of TJ proteins in human microvascular ECs [211]. Interestingly, blocking VEGF resulted in decreased edema formation and injury following ischemia [212]. Finally, ET-1 is a potent vasoconstrictor that is implicated in poorer outcomes following brain insults, and it binds to endothelial-cell-specific ET_B_ receptors. Enhanced expression occurs as early as 4 h following TBI [143]. Over-expression of ET-1 in astrocytes increases vasogenic edema, vasospasms, and reactive gliosis [142,144]. Intriguingly, administration of an ET_B_ antagonist improved BBB permeability and edema following traumatic brain injury in correlation with decreased expression of MMP-9 and VEGF-A, indicating a potential upstream mechanism of BBB breakdown by these molecules [145]. These findings highlight a greater need to evaluate the mechanisms driving vascular–astrocyte crosstalk and its influence over BBB function following TBI.

### 3.3. Endothelial-Derived Influences on the BBB Niche

Endothelial cells interact with perivascular cells in numerous ways to regulate the BBB. Endothelial intracellular signaling is modulated through direct mechanical injury and through activation of receptors or transmembrane proteins such as ET_B,_ Ephs, ICAM, and Mfsd2a [146,150,151,213]. Endothelial-specific ET_B_ activation via its ligand, ET-1, causes increased transendothelial transport of monocytes [146]. Early activation of the endothelium following TBI also causes up-regulation of ICAM-1, a cell adhesion molecule on endothelial cells important for leukocyte trafficking and BBB regulation [150,151]. Similarly, major facilitator superfamily domain containing 2a (Mfsd2a), a transmembrane protein that is integral to the development and maintenance of an intact BBB [214], is decreased following brain injury in conjunction with an up-regulation in vesicle trafficking proteins such as caveolin-1, and BBB disruption [152,153,154]. Over-expression of Mfsd2a following injury attenuated these effects, therefore, Mfsd2a provides protection to the BBB by reducing vesicular transcytosis [152,153,154]. These findings indicate that the endothelial cell response is a key driver of neuroinflammation and subsequent BBB disruption. Moreover, Eph receptor signaling, the largest family of receptor tyrosine kinases, has been shown to mediate secondary injury and may also influence BBB function [155,213]. Deletion of the class B receptor, EphB3, increased BBB integrity, endothelial cell survival, and enhanced astrocyte-EC interactions in mice following CCI injury [213]. Pharmacological inhibition of certain endothelial cell-specific transporters has also proven beneficial for outcome following brain injury. Suppression of the Na(+)-K(+)-2Cl(-) cotransporter, NKCC1, and the trauma-/ischemia-induced SUR1-regulated NC(Ca-ATP) (SUR1/TRPM4) channel by bumetanide and glibenclamide, respectively, show reduction of capillary failure in rats following TBI and ischemia [107]. Additionally, blocking SUR1 using glibenclamide reduces progressive secondary hemorrhage, necrotic lesion, and neurobehavioral deficits following TBI [108], as well as improved rapid learning and long-term protection in the hippocampus in rats after TBI [109]. Further exploration of the cellular and molecular mechanisms involved in BBB dysfunction will lead to improved targets for BBB therapy.

### 3.4. Age-Dependent Responses

Divergent age-at-injury effects on the BBB response has also been implicated in functional outcome following TBI. Secondary injury elicited across the age spectrum may substantially change the course of BBB disruption and prognosis for patients. A whole-blood transcriptomic profile of juvenile mice at 4 days following TBI suggests suppression of neuroinflammation through attenuation of the innate immune activation and pattern recognition receptor (PRR) signaling such as Dectin-1 compared to P60-80 adult mice [215]. Moreover, aged mice show increased MMP-9 activation concomitant with decreased BBB repair [216], as well as decreased motor function, increased edema, and prolonged opening of the BBB [217]. This disparity may be attributable to changes in AQP4 or TJ protein expression between ages, as juveniles have been reported to have delayed increases in AQP4 and preserved expression of collagen-IV, laminin, claudin-5, occludin, and ZO-1 following brain injury [218,219]. Furthermore, juvenile mice exhibited decreased BBB permeability 4 days post-TBI in correlation with decreased lesion volume, improved behavioral function, and restored cerebral blood flow [220]. Pharmacological inhibition of Tie2 receptor signaling in juvenile mice subjected to injury reversed the BBB-protective phenotype, therefore, providing a potential age-related mechanism through Tie2/angiopoietin signaling [220]. Interestingly, vascular integrity was assessed in plasma protein from human TBI patients which showed decreased Ang1 expression, as well as, Ang1/Ang2 ratio as compared to controls [221]. This pathway may represent an additional key predictive marker of vascular impairment in TBI. Human studies that further evaluate age-related changes in vascular-specific biomarkers may help expand prognostic and therapeutic approaches across the age spectrum.

### 3.5. Therapeutic Targeting of BBB Disruption

Animal models have been instrumental in understanding and testing novel treatments aimed at preventing BBB breakdown after TBI (Table 3). In a model of murine mild TBI, administration of calpain III attenuated BBB breakdown as shown using Evans blue (EB) analysis, a method that quantifies albumin extravasation in the brain. This was confirmed using intravital microscopy to analyze vascular leakage in the brain [28,222]. Similarly, diffuse injury using weight drop in mice, showed that administration of basic fibroblast growth factor (bFGF) decreased BBB permeability 24 h following injury when assessed using FITC-dextran and was associated with increased expression and colocalization of ZO-1, claudin-5, and occludin with the vessel marker platelet/endothelial cell adhesion molecule-1 (PECAM-1 or CD31) [29,222]. Catechin treatment has been another promising therapy as studies have reported decreased water accumulation, decreased inflammatory markers, and increased BBB integrity through expression of TJ proteins in rat models of TBI [31,222]. Intravenous adrenomedullin, an endogenous peptide that plays a role in BBB integrity, is also a promising therapeutic target because it exhibits anti-inflammatory and anti-apoptotic properties following fluid percussion injury (FPI) in rats by decreasing TNF, IL-1β, and IL-6 levels and brain edema, and by increasing BBB stability [32]. Additionally, combination treatment with valproic acid and fresh-frozen plasma following CCI and hemorrhagic shock in swine resulted in increased TJ proteins, ZO-1, and claudin-5, along with increased laminin protein in the extracellular matrix [34]. Further targeting of tight-junction protein expression is seen in novel studies addressing miRNA-mediated therapy following brain injury [223,224]. In rats, infusion with miR-21 agomir following TBI improved neurological outcome, activation of the Ang1/Tie2 axis, and subsequent promotion of tight junction expression [223]. Likewise, administration of miR-501-3p following chronic cerebral hypoperfusion was shown to prevent the loss of tight junction, ZO-1 [224]. These studies suggest miRNA modulation may represent a novel treatment strategy for preventing disruption of the BBB following brain injury.

Targeting BBB-related astrocytic responses using therapeutics is also an encouraging avenue. Treatment with acetazolamide, an FDA-approved drug already administered for disorders such as glaucoma, epilepsy, and heart failure, was shown to prevent redistribution of AQP4 and concurrently diminish cytotoxic edema in a murine model of TBI [208]. AQP4 modulators represent a favorable approach to reducing BBB disruption and secondary injury following TBI. However, aquaporins have many isoforms in humans, which could lead to off-target effects. Their structure also leads to poor druggability and discrepancies in screening assays. Unfortunately, highly-specific modulators of AQP4 have yet to be developed [227,228]. Lastly, targeting the Ang/Tie2 axis may be an attractive target for BBB therapy. Although not assessed in TBI, over-expression of Ang1 following ischemia resulted in decreased infarct volume and increased expression of TJ proteins [225,226]. Suppression of Tie2 function also was shown to attenuate BBB protection in a juvenile model of TBI [220] suggesting that promoting Tie2 activation may represent a strategy for limiting BBB breakdown. These and other mechanistic targets are described in Table 3.

## 4. Discussion

Despite continued advancements in the characterization and understanding of secondary injury responses following TBI, there is a growing need for advanced assessments of BBB function. A novel tool for assessing immediate changes may include the development of a “BBB on a chip”. This technique would mimic the physiology and function of the BBB under different pathological states such as neuroinflammation or hypoxia. Additionally, enhanced BBB chip models have been developed that recapitulate receptor-mediated transcytosis across the barrier, which could provide insights into disruption caused by increased trafficking, as well as, an innovative analysis into drug delivery [229,230]. Another unresolved issue is rapid assessment of BBB breakdown following insult. Considerations should be made into advancing neuroimaging, as well as potential intravenous tracers to define more precise measurements and locations of BBB disruption. The current accepted tracer used in MRI detection of injury severity is gadolinium; however, incorporating other tracers similar to those used in animal models may provide greater assessments of BBB breakdown in the brain parenchyma. Furthermore, exploration into the chronic effects of BBB dysfunction may aid our understanding of comorbidities associated with TBI such as Alzheimer’s disease, post-traumatic epilepsy, and chronic traumatic encephalopathy. Pre-clinical and clinical models of TBI will be paramount in future studies to address cellular and molecular changes within the BBB niche in order to advance discovery of biomarkers for early, non-invasive detection in human patients.

## Figures and Tables

**Table 1 ijms-21-03344-t001:** Models and Mechanisms of Blood–Brain Barrier (BBB) Disruption following Traumatic Brain Injury (TBI).

	Species	Method of Evaluation and Model	Timepoint	Major Findings	Reference
**Pre-clinical**	**Mouse**	Evans blue (EB) extravasation-controlled cortical impact (CCI) EB extravasation and intravital microscopy-mild TBI (mTBI) EB Extravasation and FITC-dextran-weight drop TBI	1–21 days post injury (dpi) 1 and 3 dpi * 60 min * 1 dpi *	Apolipoprotein E4 (Apoe4) impairs BBB repair through decreased pericyte and tight junction (TJ) expression with increased matrix metalloproteinase-9 (MMP-9) Calpain III administration before or after decreases BBB permeability Basic fibroblast growth factor (bFGF) given intranasally prior to TBI decreased BBB permeability and increased expression and colocalization of TJ proteins	[27,28,29]
	**Rat**	Anti-IgG stain for extravasation-CCI EB extravasation-CCI BBB permeability-FPI	1 *, 3 *, 7, 60, 180 dpi 1 dpi * 2 *, 3 *, and 7 * dpi	Decreased expression of TJ proteins Catechin administered via oral gavage decreased BBB leakiness, swelling, and inflammation Adrenomedullin treatment following TBI decreased BBB permeability and increased aquaporin 4 (AQP4) expression	[30,31,32]
	**Pig**	Evans blue albumin extravasation-severe TBI Serum protein fibrinogen (FBG) IHC-concussion Immunofluoroscopic evaluation-CCI	6 h * 6–72 h * 6 h *	Exosome treatment 1 h following CCI decreased BBB permeability and increased TJ protein levels BBB disruption overlap with axonal injury pathology, similarities to human Valproic acid and fresh-frozen plasma combination treatment following TBI increased TJ expression	[22,33,34]
**Clinical**	**Acute phase**	Serum protein fibrinogen (FBG) IHC—severe TBI (post-mortem) FBG immunoreactivity and IgG IHC Serum S100b and MRI-DTI-sub-concussive events Serum S100b—severe TBI Serum UCHL1-moderate to severe TBI Plasma miRNA-Severe TBI	6–72 h * Survival 10 h – 13 dpi * Immediately following game, 6 months for DTI 12 h * 12 h * 24 h	FBG extravasation but no axonal injury parallels Multifocal FBG extravasation and IgG immunoreactivity Increased serum S100b indicating BBB disruption and persistent abnormalities on magnetic resonance imaging-diffusion tensor imaging (MRI-DTI) Increased S100b levels indicating BBB disruption Increased serum Ubiquitin C-Terminal Hydrolase L1 (UCHL1) Alterations in expression	[19,33,35,36,37,38]
	**Long-term** post-mortem	FBG immunoreactivity and IgG IHC	Survival 1–47 years post-injury	Multifocal FBG extravasation and IgG immunoreactivity	[19]

(* indicates significant difference from control).

**Table 2 ijms-21-03344-t002:** Cellular and Molecular Mechanisms of BBB Breakdown following TBI.

	Expression	Origin/Cell Type(s)	Findings	Reference
Vasogenic edema	N/A	Endothelial cells	Increased transendothelial extravasation of serum proteins Increased vascular endothelial growth factor (VEGF) Increased intracranial pressure (ICP) and water accumulation in extracellular space Increased BBB breakdown	[41,42,45,101,102,103]
Cytotoxic edema	N/A	Astrocytes, endothelial cells, neurons	Astrocytic AQP4 redistribution Sur1/Trpm4 upregulated after injury in endothelial cells (ECs), neurons, and astrocytes Increased NKCC1 expression in neurons and glia Increased ICP and BBB breakdown Increased water accumulation in cells	[42,45,104,105,106,107,108,109]
Caveolae	Increased	Endothelium	Albumin extravasation Increased neuroinflammation Plasma Protein Influx Breakdown TJ proteins Increased transcellular permeability	[49,110]
Albumin	Increased presence in brain	Peripheral blood	Glial activation: Microglial and astrocyte release of chemokine, cytokines and MMPs Further BBB breakdown	[110,111,112,113,114]
Rho/Rock, PKC, MAPK pathways	Increased downstream activation	Glial cells	Breakdown of TJ proteins Increased Cytokine signaling Increased BBB breakdown	[115,116,117,118,119,120]
Oncostatin M	Increased following TBI	Monocytes	Increased IL-6 and ERK 1/2 expression in astrocytes Increased prostaglandin E2 and cyclooxygenase-2 in astrocytes Activation of glial and endothelial cells leading to pro-inflammation BBB breakdown	[121,122,123,124]
Reactive oxygen species (ROS)	Increased following mitochondrial damage	Astrocytes, microglia, ECs, and neurons	Glial activation Glial release of MMPs, IL-6, IL-1 Increase in ICAM-1, leukocyte adhesion and migration BBB breakdown	[125,126,127,128,129,130,131]
Aquaporin-4 (AQP4)	Varied and redistributed	Astrocytic end-feet	Vasogenic and cytotoxic edema Decreased expression leads to BBB breakdown Redistribution from end-feet to other membranes Disrupted transport and clearance of water/ion leading to accumulation	[101,102,105,132,133,134,135,136,137,138]
MMPs	Increased	Astrocytes	Breakdown of TJ proteins, occludin and claudin-5 Breakdown BBB	[103,139]
Vascular endothelial growth factor (VEGF)	Increased	Monocytes Astrocytes (VEGF-A)	Reduced claudin-5 and occludin expression Interaction with TYMP, represses TJ expression Induces pro-inflammation BBB breakdown	[140,141]
Endothelin signaling	Potent vasoconstrictor-upregulated	Astrocytes Endothelial cells (ET_B_ receptor)	Increased transendothelial transport of monocytes (ET_B_) Increased vasogenic edema and vasospasms Glial activation BBB breakdown	[142,143,144,145,146]
Glutamate	Increased accumulation	Astrocytes	Decreased expression of glutamate transporters EAAT1 and EAAT2 Activation of N-methyl-D-aspartate (NMDA) receptors and excitotoxicity in neurons and endothelial cells Increased vascular permeability	[125,147,148,149]
Immune cell adhesion molecule-1 (ICAM-1)	increased	Endothelial cells	Increase leukocyte trafficking BBB breakdown	[150,151]
Major Facilitator Superfamily Domain Containing 2A (Mfsd2a)	Decreased	Endothelial cells	Increased vesicle trafficking and transcytosis BBB breakdown	[152,153,154]
Eph/Ephrin signaling	Increased	Endothelial cells, Astrocytes pericytes, microglia, immune cells	Decreased TJ protein, zona occludens Increased neuroinflammation BBB breakdown	[155]

**Table 3 ijms-21-03344-t003:** Genetic and Pharmacological Approaches to BBB Modification.

	Pharmacological/Genetic Modification	Origin/ Cell Type(s)	Findings	Reference
Ang1	Over-expression (adeno-associated and lentivirus vector)	Glial Immune cells	Up-regulation of TJ protein, claudin-5, occludin, and ZO-1 Reduced infarct volume	[225,226]
Tie2	Pharmacological inhibition (soluble Tie2 inhibitor)	Endothelial cell Subset immune cells	Reduced occludin expression in endothelial cells Increased VEGF in endothelial cells Attenuated BBB breakdown and neuroprotection in juvenile mice	[220]
Calpain III	Pharmacological inhibition	Ubiquitously	Improved BBB integrity Displaced ZO-1	[28]
Basic fibroblast growth factor (bFGF)	Pharmacological over-expression	Neural stem cells Capillary endothelial cells Vascularized tissue (tumors)	Increased colocalization of ZO-1, claudin-5, and occludin Improved BBB integrity	[222,29]
Catechin	Pharmacological administration of tea flavonoid, antioxidant	High affinity binding to laminin receptors Potentiate brain derived neurotrophic factor (BDNF)	Decreased water accumulation Decreased inflammation Increased expression of TJ proteins Increased BBB integrity	[31]
EphB3	Genetic knockdown	Astrocytes	Increased pericyte-EC interactions Increased astrocyte-EC interactions Increased endothelial cell survival Increased BBB integrity	[213]
ET_B_	Antagonist administration	Endothelial cells Astrocytes	Increased anti-inflammatory response Decreased transendothelial passage of monocytes	[145]
miRNAs	Agomir administration (i.e., miR-21, miR-501-3p) Antagomir administration (i.e., miR-21 antagomir)	N/A	Activation of Ang1/Tie2 axis, expression of TJ proteins Suppressed TNF, increased expression of ZO-1 Neuroprotection Increased BBB stability	[223,224]
AQP4	AQP4 inhibitor: acetazolamide Inhibition by CCL2/p-38 MAPK inhibitor	Astrocytes	Eliminated cytotoxic edema Reduced edema Prevented AQP4 redistribution	[208,120]

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
