# Peer review of "Mechanisms of Blood–Brain Barrier Dysfunction in Traumatic Brain Injury"

_ijms, 2020, doi:10.3390/ijms21093344_

Round 1
Reviewer 1 Report
In this manuscript, authors reviewed mechanisms and detection methods of blood-brain barrier dysfunction in experimental and clinical traumatic brain injury, and suggested that therapeutic targeting of the BBB disruption is a novel strategy for TBI. Although this manuscript contains well compiled information and interesting observations, there are several points to be improved.
1) In Table 1, “Time point” is difficult to understand because days and dpi are mixed. Please revise more easy to understand.
2) MMP-9 is also a key player for vasogenic edema. Please added to MMP-9 in the column of vasogenic edema in Table 2.
3) In Table 2, the columns of vasogenic edema and cytotoxic edema may be separated to other columns. As a Table 3, “Cellular and Molecular Mechanisms of BBB Breakdown following TBI” may be shown. Moreover, “Effects” and “Implications” may be integrated.
4) In “4.5. Therapeutic Targeting of BBB Disruption”, the therapeutic candidate drugs including calpain and bFGF may be summarized with expected protective mechanisms as a “Table 4”. Moreover, microRNAs and endothelin receptor antagonist may be also therapeutic candidates for the TBI-induced BBB disruption.
Author Response
We would like to thank the reviewers for providing constructive and thoughtful comments. Overall, the points raised were insightful and greatly appreciated. The revised manuscript has addressed the comments through clarification in the writing and/or presentation of the tables, which we believe strengthen the overall work and highlights of the review. We hope these modifications to our manuscript are satisfactory for the reviewers.
Reviewer #1:
- In Table 1, “Time point” is difficult to understand because days and dpi are mixed. Please revise more easy to understand.
- We thank the reviewer for pointing this out and have made consistent the days throughout the table as “dpi”.
- MMP-9 is also a key player for vasogenic edema. Please added to MMP-9 in the column of vasogenic edema in Table 2.
- We thank the reviewer for noting the importance of MMP-9. We have added MMP-9 under vasogenic edema and included information in the body of the text pages 335-337.
- In Table 2, the columns of vasogenic edema and cytotoxic edema may be separated to other columns. As a Table 3, “Cellular and Molecular Mechanisms of BBB Breakdown following TBI” may be shown. Moreover, “Effects” and “Implications” may be integrated.
- We thank the reviewer for this suggestion. We have separate these forms of edema into separate rows in Table 2. We have also included a table 3 for therapeutics and combined effects and implications into one column of “findings”
- In “4.5. Therapeutic Targeting of BBB Disruption”, the therapeutic candidate drugs including calpain and bFGF may be summarized with expected protective mechanisms as a “Table 4”. Moreover, microRNAs and endothelin receptor antagonist may be also therapeutic candidates for the TBI-induced BBB disruption.
- We thank the reviewer for this suggestion. We have created a new Table 3 for therapeutic targeting of BBB and included these candidates mentioned.
Reviewer 2 Report
Dear Editor
The manuscript by Cash and Theus regarding the BBB dysfunction in TBI is interesting and informative. This topic is important and of general interest. The review is comprehensive, up to date and covers a good range of relevant literature. Authors were successful in providing well compiled opinions and summaries. The provided tables will be a good source for IJMS readers.
However, there is a number of major and minor points that would need to be addressed in order to improve the quality of this paper before it can be accepted for publication:
General:
-Authors need to define abbreviations whenever they appear first in the manuscript and use them throughout. Examples: TBI in line 60 and BBB in line 80.
-Authors have abbreviated aquaporin 4 to AQ4. It’s rather unusual term since they have been usually referred to as AQP4. This needs to be corrected throughout the manuscript.
-It’s odd that authors have repeated the exact same abstract at what it supposed to be the start of their general introduction (line 25). Authors need to come with a general overview of their review and what’s new in it before starting with TBI (line 35).
Other points:
-Line 115: discuss the ethical consideration of using big animal models in addition to cost and space.
-Line 135: authors need to mention some of the recent advances in diffusion tensor and multiple echo time arterial spin labelling MRI for TBI studies:
https://www.ncbi.nlm.nih.gov/pubmed/30557661
https://www.ncbi.nlm.nih.gov/pubmed/30063207
-Table 2: a number of recent studies have highlighted the role of rapid AQP4 translocation rather than expression in astrocytes swelling during the acute phase of cytotoxic edema. Authors need to update the section of “cytotoxic edema” with these references:
-https://www.ncbi.nlm.nih.gov/pubmed/26013827
https://www.ncbi.nlm.nih.gov/pubmed/31242419
-Table 2 MAPK and its related paragraph in the main text: MAPK pathway is known to be involved in regulating various AQPs. Authors should mention a couple of different examples for its known roles in various cells and human samples in health and disease conditions. References to be included:
Cells:
https://www.ncbi.nlm.nih.gov/pubmed/22886825
https://www.ncbi.nlm.nih.gov/pubmed/22676888
Human and murine samples:
https://www.ncbi.nlm.nih.gov/pubmed/28715131
https://www.ncbi.nlm.nih.gov/pubmed/12944406
-Line 251: authors touched on an important point regarding the oxidative stress as a disrupter of BBB. This needs to be discussed in the light of changing brain energetics as well. References to be included:
https://www.ncbi.nlm.nih.gov/pubmed/31318452
https://www.ncbi.nlm.nih.gov/pmc/articles/PMC3268209/
-Line 268: AQPs are historically known to be passive transporters of water. Lines of evidence in the last decade have highlighted the diverse function of AQPs beyond water homeostasis. Authors need to cover this point. A reference to be included:
https://www.ncbi.nlm.nih.gov/pubmed/26365508
Moreover, a subgroup of AQP water channels also facilitates transmembrane diffusion of small, polar solutes not only water; aquaglyceroporin. References:
https://www.ncbi.nlm.nih.gov/pubmed/16715408
https://www.ncbi.nlm.nih.gov/pubmed/31889130
-Line 271: authors have successfully highlighted the mixed roles of AQP4 in cytotoxic and vasogenic edema. AQP4 has been validated as an important drug target but there is no single drug that has yet been approved to successfully target it. This needs to be mentioned, references:
https://www.ncbi.nlm.nih.gov/pmc/articles/PMC4067137/
https://www.ncbi.nlm.nih.gov/pmc/articles/PMC6480248/
-Line 277: The increased AQP4 expression and the redistribution/surface localization can be two different concepts. Previous studies have shown an increased in AQP4 membrane localisation in primary human astrocytes which wasn’t accompanied by a change in AQP4 protein expression levels. Reference:
https://www.ncbi.nlm.nih.gov/pmc/articles/PMC5765450/
-Line 312: Genetic ablation of Mfsd2a results in a leaky BBB from embryonic stages through to adulthood, but the normal patterning of vascular networks is maintained. Electron microscopy examination reveals a dramatic increase in CNS-endothelial-cell vesicular transcytosis in Mfsd2a(-/-) mice, without obvious tight-junction defects. Authors need to discuss this point at the Endothelial-derived influences on BBB. Reference to be included:
https://www.ncbi.nlm.nih.gov/pubmed/24828040
-BBB-on-chip is a new exciting concept to study a humanized model of BBB. Authors have briefly touch on this point in the discussion. It might be a good idea to expand on it a little bit more. Suggested references:
https://www.ncbi.nlm.nih.gov/pubmed/30125269
https://www.ncbi.nlm.nih.gov/pubmed/31197168
https://www.ncbi.nlm.nih.gov/pmc/articles/PMC6117964/
Best
Author Response
Authors need to define abbreviations whenever they appear first in the manuscript and use them throughout. Examples: TBI in line 60 and BBB in line 80.
- We thank the reviewer for noting this. We have reviewed and corrected all abbreviations as defined when they first appear in the manuscript.
- Authors have abbreviated aquaporin 4 to AQ4. It’s rather unusual term since they have been usually referred to as AQP4. This needs to be corrected throughout the manuscript.
- We thank the reviewer for noting this, AQP4 is now used throughout.
- Line 115: discuss the ethical consideration of using big animal models in addition to cost and space.
- We thank the reviewer for this suggestion; we have added ethical consideration as a constraint to large animal research. We did not expand on this, however, as the review is intended to focus solely on the mechanisms of the BBB.
- -Line 135: authors need to mention some of the recent advances in diffusion tensor and multiple echo time arterial spin labelling MRI for TBI studies:
- We have include the possibility of using arterial spin labeling for BBB in TBI.
- Table 2: a number of recent studies have highlighted the role of rapid AQP4 translocation rather than expression in astrocytes swelling during the acute phase of cytotoxic edema. Authors need to update the section of “cytotoxic edema” with these references:
- We thank the reviewer for pointing this out. We agree these studies need mentioned and have updated table 2.
- Table 2 MAPK and its related paragraph in the main text: MAPK pathway is known to be involved in regulating various AQPs. Authors should mention a couple of different examples for its known roles in various cells and human samples in health and disease conditions.
- We thank the reviewer for noting this and have included the references for these roles on pages 265-272.
- Line 251: authors touched on an important point regarding the oxidative stress as a disrupter of BBB. This needs to be discussed in the light of changing brain energetics as well.
- We thank the reviewer for this suggestion. Our referenced materials stay limited/focused on those related to the BBB and are noted in the text of the manuscript.
- Line 268: AQPs are historically known to be passive transporters of water. Lines of evidence in the last decade have highlighted the diverse function of AQPs beyond water homeostasis. Authors need to cover this point. A reference to be included:
- We thank the reviewer for raising this important point. We have included this reference under vascular-astrocyte coupling.
- Line 271: authors have successfully highlighted the mixed roles of AQP4 in cytotoxic and vasogenic edema. AQP4 has been validated as an important drug target but there is no single drug that has yet been approved to successfully target it. This needs to be mentioned, references:
- We have added the suggested references to therapeutic approaches section
- Line 277: The increased AQP4 expression and the redistribution/surface localization can be two different concepts. Previous studies have shown an increased in AQP4 membrane localisation in primary human astrocytes which wasn’t accompanied by a change in AQP4 protein expression levels. Reference:
- We agree this is an important point to make. We have added this into the section on AQ4 under vascular-astrocyte coupling.
- Line 312: Genetic ablation of Mfsd2a results in a leaky BBB from embryonic stages through to adulthood, but the normal patterning of vascular networks is maintained. Electron microscopy examination reveals a dramatic increase in CNS-endothelial-cell vesicular transcytosis in Mfsd2a(-/-) mice, without obvious tight-junction defects. Authors need to discuss this point at the Endothelial-derived influences on BBB.
- The review raises an important mechanism we overlooked and are grateful for the comment. This pathway in the regulation specifically in brain injury was included under this section.
- BBB-on-chip is a new exciting concept to study a humanized model of BBB. Authors have briefly touch on this point in the discussion. It might be a good idea to expand on it a little bit more. Suggested references:
- We thank the reviewer for highlighting this important model. We agree with the need to expand on this discussion and have therefore provided such content on pages 444-447.
Round 2
Reviewer 2 Report
Dear editor
I would like to thank the authors for their efforts to revise the manuscript in the light of the raised concerns and suggestions. All my comments have been addressed by the authors accordingly.
The newly added sections, table 3 and updated references have helped towards the improvement of the current version compared to their earlier submission.
I would like to recommend this manuscript for publication at IJMS.
Best